# Development and Validation of an HPLC Method for the Quantitative Analysis of Bromophenolic Compounds in the Red Alga *Vertebrata lanosa*

**DOI:** 10.3390/md17120675

**Published:** 2019-11-29

**Authors:** Stefanie Hofer, Anja Hartmann, Maria Orfanoudaki, Hieu Nguyen Ngoc, Markus Nagl, Ulf Karsten, Svenja Heesch, Markus Ganzera

**Affiliations:** 1Department of Pharmacognosy, University of Innsbruck, Innrain 80-82, 6020 Innsbruck, Austria; stefanie.hofer@uibk.ac.at (S.H.); maria.orfanoudaki@uibk.ac.at (M.O.); markus.ganzera@uibk.ac.at (M.G.); 2Institute of Hygiene and Medical Microbiology, Medical University of Innsbruck, Schöpfstraße 41, 6020 Innsbruck, Austria; m.nagl@i-med.ac.at; 3Institute of Biological Sciences, Applied Ecology & Phycology, University of Rostock, Albert-Einstein-Str. 3, 18059 Rostock, Germany; ulf.karsten@uni-rostock.de (U.K.); svenja.heesch@uni-rostock.de (S.H.)

**Keywords:** marine algae, macro algae, bromophenols, HPLC, quantification, isolation

## Abstract

Bromophenols are a class of compounds occurring in red algae that are thought to play a role in chemical protection; however, their exact function is still not fully known. In order to investigate their occurrence, pure standards of seven bromophenols were isolated from a methanolic extract of the epiphytic red alga *Vertebrata lanosa* collected in Brittany, France. The structures of all compounds were determined by NMR and MS. Among the isolated substances, one new natural product, namely, 2-amino-5-(3-(2,3-dibromo-4,5-dihydroxybenzyl)ureido)pentanoic acid was identified. An HPLC method for the separation of all isolated substances was developed using a Phenomenex C8(2) Luna column and a mobile phase comprising 0.05% trifluoroacetic acid in water and acetonitrile. Method validation showed that the applied procedure is selective, linear (R^2^ ≥ 0.999), precise (intra-day ≤ 6.28%, inter-day ≤ 5.21%), and accurate (with maximum displacement values of 4.93% for the high spikes, 4.80% for the medium spikes, and 4.30% for the low spikes). For all standards limits of detection (LOD) were lower than 0.04 μg/mL and limits of quantification (LOQ) lower than 0.12 μg/mL. Subsequently, the method was applied to determine the bromophenol content in *Vertebrata lanosa* samples from varying sampling sites and collection years showing values between 0.678 and 0.005 mg/g dry weight for different bromophenols with significant variations between the sampling years. Bioactivity of seven isolated bromophenols was tested in agar diffusion tests against *Staphylococcus aureus* and *Escherichia coli* bacteria. Three compounds showed a small zone of inhibition against both test organisms at a concentration of 100 µg/mL.

## 1. Introduction

Naturally occurring organobromine compounds are very unique in the plant kingdom, as bromination does not take place in terrestrial plants. Marine organisms are able to biosynthesize such substances because they have direct access to bromides naturally occurring in seawater but also by virtue of a rare enzyme called vanadium bromoperoxidase [1]. These brominated molecules are thought to be responsible for the typical sea-like taste and flavor of seafood [2]. Bromophenols as a subtype of this group can mainly be found in red algae; for example, in the genera *Odonthalia, Polysiphonia, Rytiphlaea, Vadalia,* and *Symphocladia* [3]. A broad range of pharmacologically relevant activities has been reported for this class of substances in vitro and in vivo, ranging from antioxidant, antimicrobial, antithrombotic, and antidiabetic to anticancer effects in humans [4]. Although some of these phytochemicals are thought to play a role in the chemical protection of marine organisms, their functional role is still not fully understood, as most studies on brominated molecules focus on their bioactivity rather than their ecological relevance [5,6].

Various bromophenols have been isolated and structurally identified from different red algae in the past, but little information is available on their actual content, their seasonal variation [7], and the possible influences of their sampling location and other environmental factors. Phillips and Towers (1980) developed an HPLC diode array detection (HPLC-DAD) method for the determination of different bromophenols in red algae. However, baseline separation could not be achieved for all of their standards, and therefore only one bromophenol (lanosol) was determined quantitatively [8,9]. More recently, HPLC-MS/MS and HPLC with fluorescence detection have been used for the quantification of bromophenols in other matrices such as seafood or water samples [10,11].

In our study, a fully validated HPLC method with simple and cost-efficient UV detection was used for the determination and quantification of seven bromophenolic compounds in the red alga *Vertebrata lanosa* (L.) T.A.Christensen (formerly *Polysiphonia lanosa* (L.) Tandy). This species grows as an obligate epiphyte on the brown alga *Ascophyllum nodosum* (L.) Le Jolis, which inhabits intertidal rocky shores of Northern Atlantic coasts [12]. Pure standards were isolated from the methanolic extract of a sample collected in Brittany (France) by using various chromatographic techniques followed by structural elucidation of all isolated compounds by NMR spectroscopy and MS. The isolated standards include one new bromophenolic substance and six that have previously been reported from other red algae [13,14,15,16]. Their content was determined in *Vertebrata lanosa* from different collection years and sampling sites. To investigate the hypothesis that bromophenols might protect against microbial infestations, antimicrobial activity of the seven standards was tested using agar diffusion tests.

## 2. Results

### 2.1. Structural Elucidation of Compound 1

Seven bromophenolic compounds (see Figure 1 for structures) could be isolated from *V. lanosa*; among them was one new natural product, compound **1**.

Compound **1** (Figure 2a) was obtained as a light purple amorphous powder and its purity was confirmed by HPLC-MS and NMR. The positive ESI-MS spectrum of the substance showed the typical isotopic pattern for dibrominated molecules at *m*/*z* 454/456/458 in the ratio of 1:2:1 [M + H]^+^ as shown in Figure 2b. High-resolution ESI-MS data ([M + H]^+^ = 455.9618) corresponded to the molecular formula C_13_H_17_^79^Br^80^BrN_3_O. The substance showed optical rotation$\;({\left[\alpha \right]}_{D}^{20}$ = –53.03 (*c* 0.7, MeOH)) and a UV λ_max_ at 216 and 292 nm.

The ^1^H NMR spectrum of **1** displayed one aromatic proton at *δ* 6.84 (s, H-6′), protons of methylene groups as two doublets at *δ*_H_ 4.28 (1H, *J* = 16.0 Hz) and 4.23 (16.0 Hz), and multiplets at *δ*_H_ 3.23, 3.11, 1.89, and 1.60. One proton resonance was also observed at *δ*_H_ 3.57 (t, 6.0 Hz). The NMR shifts of the protons and the corresponding carbons of the substructure were unambiguously assigned by ^1^H-^1^H COSY, HSQC, and HMBC experiments. The main HMBC and ^1^H-^1^H COSY correlations are shown in Figure 2a. In the HMBC spectrum, correlations from H-6′ to C-2′ and C-4′ and C-7′, in addition to signals typical for two brominated quaternary carbons (*δ*_C_ < 110 ppm) and two oxygenated carbons on the aromatic ring (*δ*_C_ < 140 ppm) revealed the substructure of a (2,3-dibromo-4,5-hydroxybenzyl)amino group. The ^1^H-^1^H COSY signals between H-2 and H-3, H-3 and H-4, and H-4 and H-5, in combination with signals of a primary amino group at C-2 (*δ*_C_ 55.88) and a carboxyl group at C-1 (*δ*_C_ 174.38), which were assigned by their HMBC correlations between H-3/H-2 and C-1, confirmed the presence of an ornithyl moiety. These two units were connected via a ureido part structure which could be determined by HMBC correlations from H-5 to C-6 and H-7′ to C-6, as well as the characteristic shift value of C-6 (*δ*_C_ 160.95). Collectively, the structure of **1** can be described as 2-amino-5-(3-(2,3-dibromo-4,5-dihydroxybenzyl)ureido)pentanoic acid; NMR shift values can be found in Table 1 and original NMR spectra of this compound in the Appendix A. It is a new natural product and we named this new bromophenol Vertebratol. The NMR data of the known bromophenolic compounds **2** to **7** (see Appendix A) were in good agreement with values from the literature [13,14,15,16].

### 2.2. HPLC Method Development

The HPLC separation of seven bromophenols contained in *Vertebrata lanosa* in less than 25 min was possible with the setup shown in Figure 3. The previously described compounds (Figure 1) were used as standards for method development. During its initial phase, four stationary phases, namely, Phenomenex Synergi MAX-RP (150 mm × 2.0 mm), Phenomenex Synergi POLAR-RP (150 mm × 4.6 mm), YMC Triart (150 mm × 3.0 mm)m and Phenomenex Luna C8(2) (150 mm × 2.0 mm,), all with comparable particle sizes (3–4 μm), were tested with two different mobile phase systems, i.e., methanol/water and acetonitrile/water. It was observed that the latter column yielded the overall best results together with acetonitrile/water, not only concerning separation of the standards but also for the analysis of the crude *V. lanosa* extract to avoid interference with other naturally occurring substances. The addition of 0.05% trifluoroacetic acid to the mobile phase was advantageous, whereas using 0.1% phosphoric acid or a 10 mM ammonium acetate buffer decreased the quality of separation. The optimum temperature was found to be 30 °C, and the elution gradient was carefully optimized. Ideal elution was performed at a flow rate of 0.25 mL/min by starting at a concentration of 2% B and rapidly increasing it to 20% within only 0.1 min, followed by an increase to 50% B in 15 min and to 70% B in another 20 min. The acetonitrile content was then raised to 98% B in a further 5 min; this composition was maintained for an additional 10 min, leading to a total runtime of 55 min. Of note, changes in the starting conditions in particular (e.g., leaving the concentration at 2% B for longer than 0.1 min (see Appendix A) or immediately starting the gradient at 20% B (see Appendix A) had a negative impact on the separation of peaks **2** and **3**. During analysis the DAD was set to 210 nm and the injection volume was adjusted to 5 μL.

### 2.3. Method Validation

Following assay development, the method was validated. Selectivity was deduced from consistent UV spectra within the peaks of interest (by using the peak purity function available in the software) and no signs of co-eluting compounds, visible by e.g., peak shoulders, were observed. The slightly wider peak of compound **2** (Methylrhodomelol) in the extract was compared to that of the corresponding standard, which was found to be highly pure based on HPLC, HPLC-MS, and NMR experiments. Additionally, the standard of Methylrhodomelol was eluted as a broader peak, indicating that this was a substance specific effect not caused by impurity. Calibration curves were established for all of the seven standards. Coefficients of determination were always 0.999 or higher in a concentration range of 102 to 0.05 µg/mL for compound **1**, of 181 to 0.06 μg/mL for compound **2**, of 95 to 0.02 µg/mL for compound **7,** and of approximately 50 to 0.02 μg/mL for compounds **3**, **4**, **5**, and **6**. LOD values varied from 0.008 and 0.038 μg/mL and LOQ ranged from 0.024 to 0.116 μg/mL. All summarized calibration data can be found in Table 2. Assay precision was assured by repeatedly extracting and analyzing sample 1 (sample *Vertebrata lanosa*, Brittany, France 2018) under optimized conditions (see Table 2). Intra-day (relative standard deviation ≤ 6.28 %) and inter-day (relative standard deviation ≤ 5.21%) variance was shown to be in the acceptable range, especially as algal material always shows a certain degree of inhomogeneity, which is reflected in the slight variations observed in the samples collected at the same time and location. Accuracy was determined by spiking a defined amount of grounded, dried material with three different concentrations of each of the standards (low, medium, and high) prior to sample preparation. The observed recovery rates showed maximum displacement rates of 4.93% (**2**, high spike), confirming the validity of this parameter, too.

### 2.4. Quantitative Analysis of Samples

Three *V. lanosa* samples collected in Brittany, together with one commercial sample and specimens from Ireland and Norway (Table 3) were used for quantitative analysis. Two of the samples from Brittany were collected at the same sampling site and season but across two different years. The third sample originated from a different place but was also collected in June 2019. The compounds were assigned at 210 nm by matching retention times and UV spectra compared to the standards; additionally, LC-MS experiments were performed. For quantification, each of the samples was measured at least in triplicates with maximum relative standard deviations of 5.21% (see Table 4).

All samples contained compounds **1** to **4**. However, compounds **5** to **7** could not be found in all specimens. For example, the sample collected in Roscoff 2019 (sample 2) did not exhibit a peak corresponding to **5**, and compound **7** could only be detected in traces. The second sample from Brittany from 2019 (sample 3, collected in Saint Pol de Léon) did not show any signals for compounds **5**–**7**, and compound **4** was present only in minor amounts. On the other hand, sample 1, collected in the previous year in Brittany (Roscoff, same sampling site and season as sample 2) as well as samples 4–6 contained all of the seven isolated bromophenols. In addition, in quantitative terms the bromophenol content varied significantly. This becomes clear when comparing the quantitative results of the three samples from Brittany. Here, the main bromophenol was always compound **1**. However, its content was approximately 20 times higher in sample 1 (0.678 mg/g) compared to samples 2 (0.035 mg/g) and 3 (0.029 mg/g). When comparing the results for sample 1 with those of other origins, fluctuations were found to be even larger, e.g., reaching up to 113 times lower values for compound **1** in sample 6.

The main bromophenol in sample 4 was compound **2**, which had a concentration of 0.048 mg/g dried algal material, while in sample 5 the predominant bromophenol was compound **6** (0.445 mg/g), and in sample 6 it was compound **3** (0.259 mg/g). The overall highest concentrations of compounds **1** (0.678 mg/g), **2** (0.175 mg/g), and **5** (0.028 mg/g) were found in sample 1 (2018, Roscoff, Brittany). However, the highest contents of compounds **3** (0.313 mg/g), **4** (0.339 mg/g), **6** (0.445 mg/g), and **7** (0.062 mg/g) were observed in sample 5 (2019, Lettermore, Ireland).

In the course of our investigations 20 additional samples were analyzed with the given method (19 species of macroalgae and one lichen, see Appendix A). The results showed that the seven bromophenols could not be found in any of the other samples.

### 2.5. Antimicrobial Activity

Compounds **5**–**7** (all tested at a concentration of 100 µg/mL) showed a small zone of inhibition in agar diffusion tests, while compounds **1**–**4** (100 µg/mL) and the controls did not inhibit bacterial growth. The diameter of the zone of inhibition was 11 mm for compound **5** and 9 mm for compounds **6** and **7** against *Staphylococcus aureus* ATCC 6538. *Escherichia coli* ATCC 11229, inhibition was minimal, with a 7 mm diameter observed for compounds **5**–**7**, and again, there was no inhibition for the other bromophenols and the controls.

## 3. Discussion

While halogenated compounds commonly occur in marine organisms, especially in macroalgae, the role of bromide-containing compounds in red algae is not well understood and data regarding their occurrence are inconsistent. To better understand their functional role in the plant kingdom but also the triggers for bromophenol formation, a fully validated HPLC method for the determination and quantification of seven bromophenolic compounds in the red alga *Vertebrata lanosa* (L.) T.A.Christensen was developed which offers the potential to monitor bromophenol concentrations of larger sample sets.

To obtain the standards seven bromophenols occurring in *V. lanosa* were isolated. Among them was one new natural product, 2-amino-5-(3-(2,3-dibromo-4,5-dihydroxybenzyl)ureido)pentanoic acid (Vertebratol). In the past, two related compounds with a 3-(2,3-dibromo-4,5-dihydroxybenzyl)ureido core structure have already been isolated from two other marine red algae of the family Rhodomelaceae, namely *Rhodomela confervoides* (Hudson) P.C.Silva and *Symphocladia latiuscula* (Harvey) Yamada [17,18]; however, in contrast to those molecules, the new molecule is linked to an ornithyl moiety. Interestingly, this compound was observed to occur in very high concentrations of up to 0.678 mg/g dried algal material in some of the *V. lanosa* samples. The concentrations of all seven bromophenols were determined in the methanolic extracts of several samples of this alga with our newly developed and fully validated HPLC method, which enabled their identification and precise quantification in less than 24 min. To further verify that peaks of interest in unknown samples correspond to the known standards, the method can easily be coupled to MS; however, 0.05% trifluoroacetic acid in the mobile phase has to be replaced with 0.1% formic acid in order to avoid ion suppression. Excellent determination coefficients (R^2^ ≥ 0.999) were achieved with low LOD values of between 0.008 and 0.038 μg/mL and limits of quantification (LOQ) ranging from 0.024 to 0.116 μg/mL. Thus, the method offers the potential to provide not only qualitative but also quantitative information about the occurrence of those bromophenols in algae even if only small amounts of material are available.

It is known that bromophenols are not randomly distributed. They usually only occur in particular genera and species as their biosynthesis requires certain enzymes and thus the corresponding genetic information. Consequently, the presence of a specific set of bromophenolic compounds can be used as a taxonomic marker for the discrimination of different species [19]. In our study, six *Vertebrata lanosa* samples were measured, and, interestingly, the determined concentrations of the different bromophenols showed great differences within the samples. For example, the values for the new natural product, substance **1** (Vertebratol), varied by up to 23 times within samples from Brittany from different years and sampling sites and even by up to 115 times when compared to samples from other countries. Additionally, the concentrations of the other six bromophenols were not uniform within the different samples. In qualitative terms, samples from Norway and Ireland showed different main bromophenols compared to the samples from Brittany. These results are in accordance with the literature because it has been reported that the content of brominated compounds like Isorhodolaureol in certain red algae such as *Laurencia dendroidea* J. Agardh vary extensively from year to year, even if the collection site is the same, showing that there have to be specific triggers that enhance the formation of bromophenols in some Rhodomelacean species [19,20]. As the number of samples was limited in the current work, we cannot deduce any environmental parameters which may determine the occurrence of bromophenolic compounds in *Vertebrata lanosa*. Further research will be necessary to identify them and their functional role in the algae. However, the method presented herein might serve as a useful tool for the analysis of samples in a larger number, as it is a cost-efficient and fast approach that was moreover fully validated according to the ICH guidelines.

In the past, bromophenols have been reported to possess radical scavenging activities [13], even though their antioxidant properties have so far been only determined in vitro [21,22]. These compounds have, moreover, been attributed with deterring herbivores and epiphytes: Shoeib et al. (2006) have noted that *V. lanosa* does not seem to be grazed upon much and appears remarkably free of epiphytes, both of which the authors suggested could be due to the presence of bromophenolic compounds [7]. However, a recent study on *V. lanosa* from Norway shows that microscopic epiphytes are present all year round, while (limited) macroscopic fouling does occur, especially during late autumn [23]. While some bromophenols have been shown to inhibit competitors or deter predators, for example in marine invertebrates such as worms [24,25], conclusive studies demonstrating the deterrence of herbivores and epiphytes in vivo are still lacking for *V. lanosa* as well as for other red algae. From published bioactivity data [4], it can be hypothesized that bromophenols might potentially play a role in defense against microbial infestations. Additionally, in our study compounds **5** (3-bromo-4-(2,3-dibromo-4,5-dihydroxybenzyl)-5-methoxymethylpyrocatechol), **6** (5-((2,3-dibromo-4,5-dihydroxybenzyloxy)methyl)-3,4-dibromobenzene-1,2-diol), and **7** (2,2’,3,3’-tetrabromo-4,4’,5,5’-tetrahydroxydiphenylmethane) showed weak antimicrobial activity against *E.coli* and *S. aureus* in the micro molar concentration range, which is similar to natural but also synthetic bromophenols that have been tested in the literature [26,27]. Interestingly, only substances that possess two 2,3-dibromo-4,5-dihydroxybenzyloxyl moieties were active. Whether this aspect is really essential for activity will be studied in further investigations.

## 4. Materials and Methods

### 4.1. Samples and Reagents

The origin of all *V. lanosa* samples is given in Table 3. The algal material used to isolate the standards was morphologically and genetically identified by two of the authors (Ulf Karsten and Svenja Heesch GenBank accession number LR738856), and voucher specimens were deposited at the Institute of Pharmacy, Pharmacognosy, at the University of Innsbruck, Austria. The solvents used for analytical work were of analytical grade, and they were obtained from Merck (Darmstadt, Germany). HPLC grade water was prepared by a Satorius arium 611 UV water purification system (Göttingen, Germany).

### 4.2. Isolation of Bromophenols from Vertebrata Lanosa

The methanol soluble fraction of the methanol extract of sample 1 was used for the isolation of the seven bromophenols shown in Figure 1. To achieve a targeted isolation, fractions were analyzed by HPLC-DAD-MS on an Agilent Technologies 1260 Infinity II instrument with an InfinityLab LC/MSD detector (Waldbronn, Germany) prior to further isolation steps using the optimum HPLC conditions (gradient as described in Section 2.2) on the Phenomenex C8(2) Luna column. However, for that purpose 0.05% trifluoroacetic acid had to be replaced with 0.1% formic acid in the mobile phase. A respective chromatogram of the crude extract is shown in the Appendix A. This step was useful, firstly because the brominated molecules show very characteristic isotopic patterns due to the relative abundance of the brome isotopes (see Figure 2b), and secondly because all of the isolated bromophenols exhibit two absorption maxima, one at 210 nm and a weaker one around 280–290 nm. The structures of all compounds were elucidated based on NMR and MS experiments and a comparison to literature values (if possible). MS data as well as characteristic NMR shift values can be found in the Appendix A. For the new compound **1** (Vertebratol), the original NMR spectra are shown there as well, while those of the other compounds can be provided upon request.

Air-dried and powdered *Vertebrata lanosa* material (600 g) was extracted with methanol at room temperature to obtain 70 g of crude extract. The methanol soluble part of this extract (40 g) was fractionated on a silica gel column using gradient elution (EtOAc to methanol to methanol:water 80:20), resulting in 14 fractions. Fractions 2 and 3 were combined and further fractionated on a Sephadex LH-20 column in methanol to afford 19 individual fractions. Fractions 10 and 12 were individually processed using a 4 g silica cartridge and a Reveleris^®^ X2 iES flash chromatography system (both from Büchi, Flawil, Switzerland). Elution was performed using a gradient of toluol (A) and EtOAc (B): 0–5 min: 2% B, 10 min: 30% B, 29 min: 70% B, 35 min: 100 % B, and 35–45 min: 100% B. The flow rate was set to 5 mL/min and detection was carried out at 254, 280, and 320 nm.

Flash chromatography of fraction 10 (123 mg) directly resulted in the isolation of compound **4** (9 mg) and the same procedure with fraction 12 led to the isolation of compound **3** (50 mg). Another flash chromatography fraction (fraction 7) required an additional purification step using a semi-preparative HPLC with a Synergi 4 u Polar-RP (250 mm × 10 mm, 4 μm; Phenomenex, Torrance, CA, USA) column and a Dionex UltiMate 3000 HPLC (Thermo, Waltham, MA, USA). Water (A) and methanol (B) were used as mobile phases with the following gradient: 0 min: 20% B to 96.5% B in 38 min at 45 °C. After a final purification step on a small Sephadex LH-20 column in methanol the isolation of compound **2** (16 mg) could be achieved.

The fractions 16, 17, and 18 of the first Sephadex column were subjected to semi-preparative HPLC on the same Synergi Polar-RP column using water (A) and acetonitrile (B), both containing 0.1% formic acid, as the mobile phase. The temperature was set at 30 °C and the flow rate to 2 mL/min. The applied gradient was: 0 min: 30% B, 13 min: 51.5% B, 40 min: 51.5% B, and 45 min: 98% B. This step resulted in the isolation of compounds **5** (6.6 mg), **6** (9 mg), and **7** (6 mg).

Furthermore, the initial fraction 12 (6.5 g) also showed a main peak with an isotopic pattern typical for brominated molecules. The fraction was dissolved in water and used for liquid–liquid extraction in a separatory funnel with EtOAc followed by butanol (BuOH). The BuOH fraction was then used for a further separation step with flash chromatography, applying a C-18 40 g cartridge from Büchi on the same flash chromatography system as previously mentioned. The flow rate was set to 12 mL/min and the following elution gradient (solvent A: water; solvent B: methanol) was applied: 0–5 min: 2% B, 10 min: 30% B, 35 min: 100% B, and 35–45 min: 100% B. The flash sub-fraction 13 (79 mg) was finally purified by semi-preparative HPLC using the Synergi Polar-RP (250 mm × 10 mm, 4 μm) column with water (A) and methanol (B). Isocratic elution at 30% B led to the isolation of compound **1** (30 mg).

### 4.3. Structure Elucidation

NMR experiments were performed on a Bruker Avance II 600 spectrometer (Karlsruhe, Germany) at 600.19 (1H) and 150.91 MHz (13C). The pure isolated compounds were dissolved in deuterated methanol from Euriso-Top (Saint Aubin, France). An Agilent InfinityLab LC/MSD System (Santa Clara, CA, USA) was used to measure the low-resolution mass spectra in positive mode at a capillary energy of 4000 V and nebulizer gas of 40.0 psi (dry gas), at 10.0 L/min and at a temperature of 300 °C; the recorded scan range was 100–1500 *m*/*z*. The high-resolution mass spectra were recorded on a micrOTOF-Q II mass spectrometer (Bruker-Daltonics, Bremen, Germany). The experiments were conducted in the positive as well as negative ESI mode by applying the following parameters: capillary energy, 4500 V for positive mode and 3500 V for negative mode; nebulizer gas, 6.4 psi; dry gas, 4.0 L/min at a temperature of 180 °C; and recorded scan range of 100–600 *m*/*z*.

### 4.4. Sample Preparation

The dried and finely milled algal material (0.1 g) was extracted five times with 1.5 mL of MeOH for 20 min each on an ultrasonic bath to achieve an exhaustive extraction of all compounds of interest. The supernatants were combined after centrifugation (1000× *g* for 5 min) and the solvent was evaporated.

The bromophenols were quantified at 210 nm, a wavelength where all the respective compounds show strong UV absorbance. However, when analyzing the samples, a broad and partially interfering peak was observed at min 7.6 (see Appendix A). The peak could successfully and selectively be removed by re-dissolving the crude extract in water and applying it onto a solid phase extraction cartridge (Phenomenex Strata C18-E, 55 μm, Torrance, CA, USA). The cartridge was then washed with water to elute the impurities (Appendix A), followed by an elution step with methanol to recover the enriched analytes as shown in Appendix A. After evaporation, samples were re-dissolved in 2 mL of methanol.

### 4.5. Analytical Conditions

All analytical experiments were performed on an Agilent 1100 HPLC (Waldbronn, Germany) using a Luna C8(2) column (150 mm × 2.0 mm, 3 µm particle size; Phenomenex, Torrance, CA, USA) as the stationary phase and a mobile phase comprising 0.05% trifluoroacetic acid in water (A) and acetonitrile (B).

Elution started at 2% B. This proportion was rapidly increased to 20% B in only 0.1 min, followed by an increase of up to 50% B in the first 15 min and to 70% in another 20 min. In the next 5 min elution was increased to 98% B and this composition was maintained for additional 10 min. The column was then re-equilibrated for 10 min prior to the next run. The DAD-wavelengths were adjusted to 210, 280, and 310 nm, whereas flow rate, sample injection volume, and column temperature were set at 0.25 mL/min, 5 μL, and 30 °C, respectively.

### 4.6. Method Validation

The newly developed HPLC method was validated according to ICH guidelines in order to ensure that it fulfilled regulatory standards. All validation results are summarized in Table 2. To establish calibration curves and to determine the linear range, a stock solution of all standards (approximately 2 mg, accurately weighed, per compound dissolved in 5.00 mL MeOH) was prepared. This solution was used for serial dilution in the ratio of 1:1. LOD and LOQ values were calculated according to the guidelines based on the given standard deviations of the response and the slope of calibration curves for each compound. The selectivity of the method was estimated by evaluating the photodiode array data and assessing the peak purity with the respective option in the operating software. Precision was confirmed by individual preparation of five sample solutions of sample 1 on each of three consecutive days and analyzing them. Variations in peak areas were calculated for all the relevant analytes and given for one day (intra-day precision) and for the three-day period (inter-day precision). Accuracy was assured by spiking sample 1 with three different concentrations of all the standards (high, medium, and low concentrations). Samples were then prepared as previously described and recovery rates determined as percentages of the actually present concentrations compared to the theoretically calculated ones.

### 4.7. Determination of Antimicrobial Activity

The test bacteria *Staphylococcus aureus* ATCC 6538 and *Escherichia coli* ATCC 11229 were grown on Mueller-Hinton (MH) agar plates. Overnight cultures from a single colony from these plates were grown at 37 °C in tryptic soy broth. Bacteria were spread separately on fresh MH plates with a swab. A punch with a diameter of 5 mm was made in each plate and filled with the respective test solution. Compounds **1**–**4** were dissolved in 50% methanol and compounds **5**–**7** dissolved in 50% dimethyl sulfoxide to a concentration of 1 mg/mL. These solutions were tenfold diluted in distilled water so that the final concentration of the compounds in the test solution was 100 µg/mL. Inoculated agar plates were grown at 37 °C overnight, and the zone of inhibition was observed. Controls containing 5% methanol and 5% DMSO were done in parallel.

## Figures and Tables

**Figure 1 marinedrugs-17-00675-f001:**
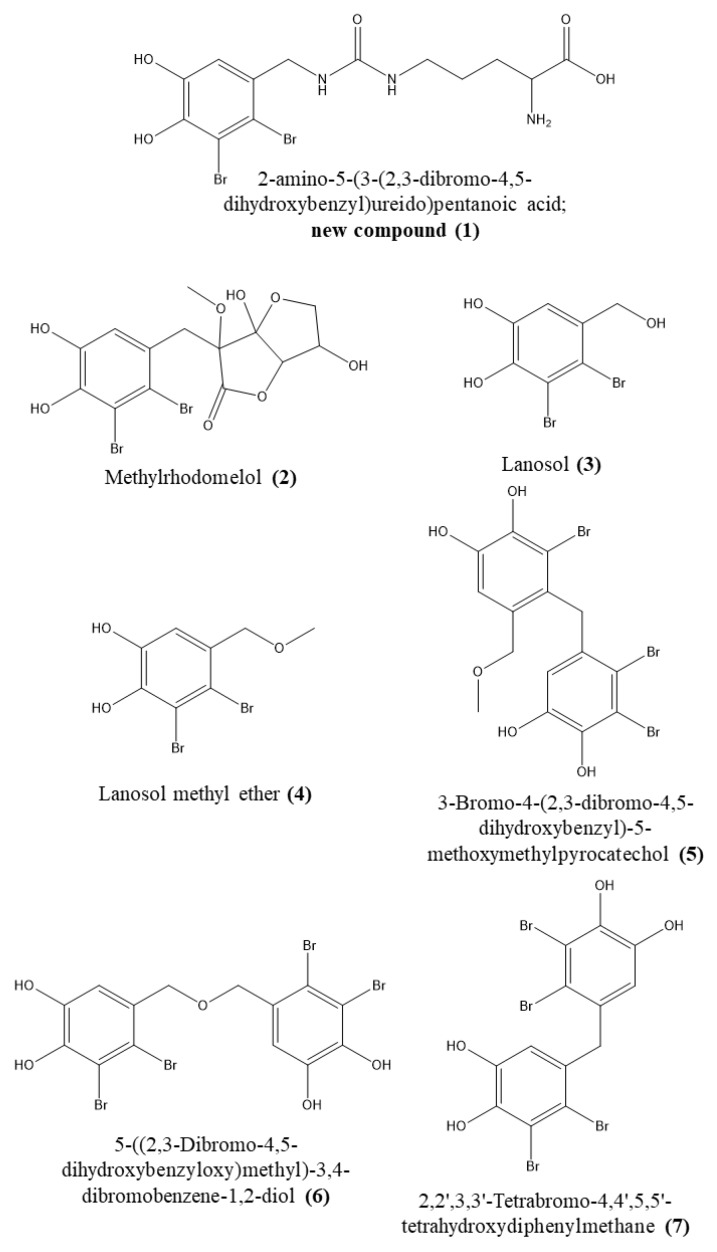
Structures of the isolated standard compounds.

**Figure 2 marinedrugs-17-00675-f002:**
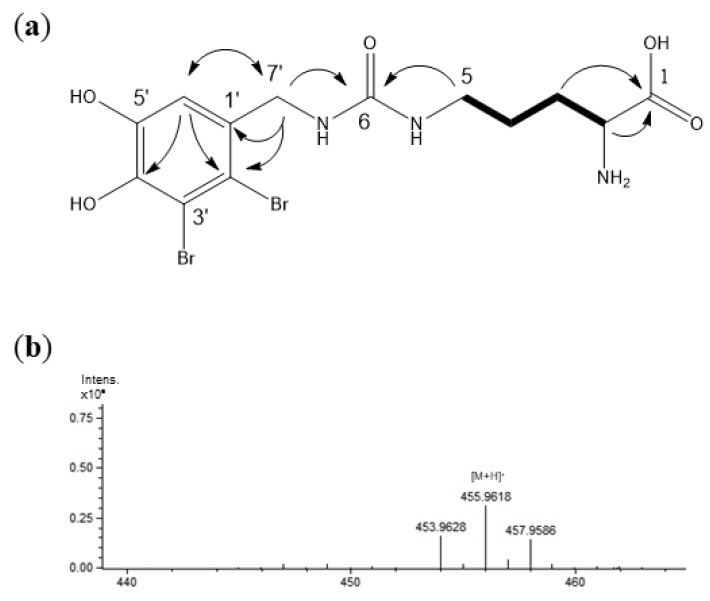
(**a**) Key HMBC correlations (H→C) and ^1^H-^1^H COSY (bold line) correlations and (**b**) high resolution ESI-MS data of compound 1 (Vertebratol).

**Figure 3 marinedrugs-17-00675-f003:**
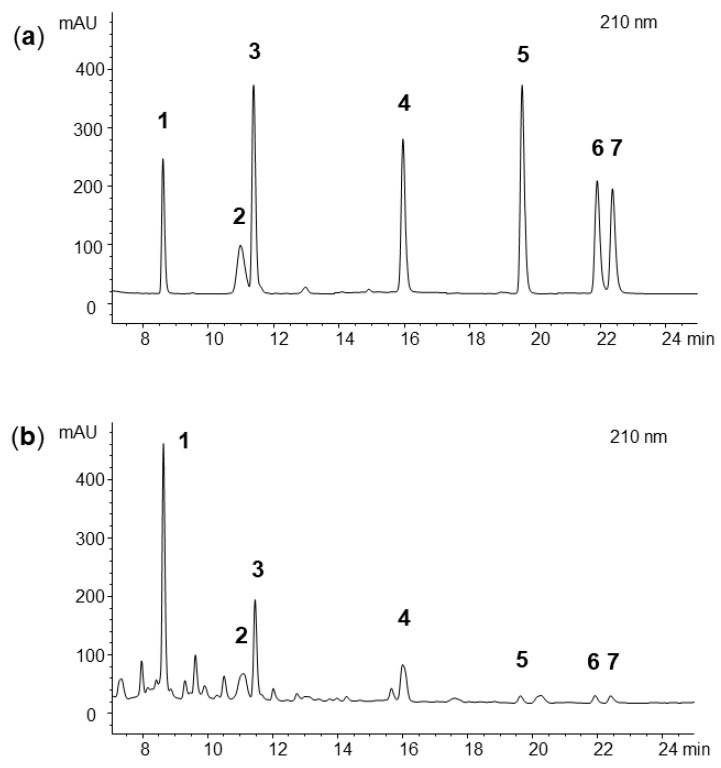
(**a**) HPLC separation of seven bromophenols under optimized conditions and (**b**) HPLC separation of a methanolic extract of *Vertebrata lanosa* (sample 1). The applied analytical conditions were column: Phenomenex Luna C8(2) (150 mm × 2.0 mm, 3 µm particle size); mobile phase: water (A) and acetonitrile (B), both containing 0.05% trifluoroacetic acid; gradient: 2% B at 0 min, 20% B at 0.1 min, 50% B at 15 min, and 70% B at 35 min; injection volume: 5 µL; flow rate: 0.25 mL/min; column temperature: 30 °C; and detection: 210 nm.

**Table 1 marinedrugs-17-00675-t001:** NMR shift values for compound **1** (Vertebratol) in MeOD; the spectra were recorded on a 600 MHz NMR instrument.

Bromophenol *m*/*z =* 455.9618 [M + H]^+^			
	^13^C	^1^H	HMBC ^a^
1	174.38, C		
2	55.87, CH	3.57, t (6.0)	1, 3, 4
3	29.40, CH_2_	1.89, m	1, 2, 4, 5
4	27.07, CH_2_	1.60, m	3, 5
5	40.19, CH_2_	3.23, m; 3.11, m	3, 4, 6
6	160.94, C		
1′	132.37, C		
2′	114.42, C		
3′	114.13, C		
4′	144.87, C		
5′	146.63, C		
6′	114.78, CH	6.84, s	2′, 4′, 7′
7′	46.010, CH_2_	4.28, d (16.0); 4.23, d (16.0)	1′, 2′, 6

^a^ HMBC correlations are stated from proton(s) to the indicated carbon.

**Table 2 marinedrugs-17-00675-t002:** Validation parameters of the HPLC method.

Calibration Data for the Seven Bromophenol Standards							
Substance	Regression Equation		Coefficient of Determination	Range (μg/mL)		LOD ^1^ (μg/mL)	LOQ ^2^ (μg/mL)
**1**	y = 83407x + 5.5502		R² = 1.0000	102–0.05		0.017	0.053
**2**	y = 98794x + 18.324		R² = 0.9999	181–0.06		0.038	0.116
**3**	y = 156713x + 34.799		R² = 0.9991	47–0.02		0.008	0.024
**4**	y = 126302x + 44.688		R² = 0.9990	49–0.02		0.011	0.034
**5**	y = 164091x + 25.645		R² = 0.9998	56–0.03		0.014	0.043
**6**	y = 126669x - 28.684		R² = 0.9995	49–0.05		0.032	0.097
**7**	y = 114941x + 42.574		R² = 0.9996	95–0.02		0.015	0.047
**Accuracy and Precision of the Assay**							
**Accuracy, High Spike (n = 3)**				**Accuracy, Medium Spike (n = 3)**			
**Substance**	**Measured Values ^3^**	**Theoretical Values ^3^**	**Displace-ment ^4^**	**Substance**	**Measured Values ^3^**	**Theoretical Values ^3^**	**Displace-ment ^4^**
**1**	89.5	97.3	2.67	**1**	69.0	71.3	3.29
**2**	135.1	142.1	4.93	**2**	49.5	51.3	3.51
**3**	41.1	41.6	1.23	**3**	28.1	26.8	4.68
**4**	48.3	49.5	2.45	**4**	27.9	27.3	1.96
**5**	49.8	51.7	3.60	**5**	29.8	28.5	4.80
**6**	43.9	45.2	2.85	**6**	25.3	25.1	1.12
**7**	87.9	87.7	0.24	**7**	46.0	45.0	2.15
**Accuracy, Low Spike (n = 3) ^3^**				**Precision (n = 5) ^5^**			
**Substance**	**Measured Values ^3^**	**Theoretical Values ^3^**	**Displace-ment ^4^**	**Day 1**	**Day 2 **	**Day 3**	**Inter-day**
**1**	50.7	51.4	1.33	1.83	0.71	1.21	2.33
**2**	29.1	30.4	4.07	1.26	0.76	1.16	2.35
**3**	12.5	13.0	3.57	2.15	0.62	1.06	2.93
**4**	9.7	10.1	4.30	2.59	1.71	3.86	2.93
**5**	6.9	7.0	0.54	3.83	6.28	2.56	5.21
**6**	5.8	5.6	3.56	1.23	0.84	2.87	1.74
**7**	23.6	23.5	0.51	1.23	0.43	2.28	2.11

^1^ LOD: limit of detection determined with the purified standards. ^2^ LOQ: limit of quantification determined with the purified standards. ^3^ Values in µg/mL. ^4^ Displacement in percent (sample *Vertebrata lanosa*, Brittany, 2018). ^5^ Maximum relative standard deviation based on the peak area given in percent.

**Table 3 marinedrugs-17-00675-t003:** Origin of analyzed *Vertebrata lanosa* samples.

Sample	Collection Site and Date
1	2018, Roscoff, Brittany, 48.727559° N; 3.987924° W, host alga *Ascophyllum nodosum*, collected and identified by U. Karsten and S. Heesch, University of Rostock
2	2019, Roscoff, Brittany, 48.727559° N; 3.987924° W, host alga *Ascophyllum nodosum*, collected and identified by S. Hofer, University of Innsbruck
3	2019, Saint Pol de Léon, 48.676896° N; 3.966533° W, host alga *Ascophyllum nodosum*, Brittany, collected and identified by S. Hofer, University of Innsbruck
4	Commercially available; Arctic Algae AS, Bodø, Norway
5	2019, Lettermore, Ireland, 53.298468° N; 9.712704° W, host alga *Ascophyllum nodosum*, collected and identified by R. Bermejo, National University of Ireland, Galway
6	2019, Bukken, Raunefjorde, Norway, 60.241183° N, 05.20225° W, host alga *Ascophyllum nodosum*, collected and identified byK. Sjøtun, University of Bergen

**Table 4 marinedrugs-17-00675-t004:** Determination of bromophenol content in six different *Vertebrata lanosa* samples.

Content of Bromophenols						
	Sample 1		Sample 2		Sample 3	
Substance	Content (µg/mg Dried Algal Material)	Relative Standard Deviation (%)	Content (µg/mg Dried Algal Material)	Relative Standard Deviation (%)	Content (µg/mg Dried Algal Material)	Relative Standard Deviation (%)
**1**	**0.678** (*)	2.33	**0.035**	0.55	**0.029**	4.52
**2**	0.175 (*)	2.35	0.027	0.64	0.013	4.08
**3**	0.158	2.93	0.032	0.66	0.008	4.18
**4**	0.094	2.93	0.015	0.68	T **^1^**	-
**5**	0.028 (*)	5.21	-	-	-	-
**6**	0.029	1.74	0.007	2.75	-	-
**7**	0.030	2.11	T **^1^**	-	-	-
	**Sample 4**		**Sample 5**		**Sample 6**	
**Substance**	**Content (µg/mg Dried Algal Material)**	**Relative Standard Deviation (%)**	**Content (µg/mg Dried Algal Material)**	**Relative Standard Deviation (%)**	**Content (µg/mg Dried Algal Material)**	**Relative Standard Deviation (%)**
**1**	0.045	0.41	0.025	0.67	0.006	2.85
**2**	**0.048**	2.66	0.140	0.36	0.105	0.30
**3**	0.035	0.40	0.313 (*)	0.76	**0.259**	0.64
**4**	0.014	1.99	0.339 (*)	0.51	0.059	0.46
**5**	0.014	1.37	0.006	2.05	0.005	1.24
**6**	0.011	1.91	**0.445** (*)	0.43	0.052	0.29
**7**	T**^1^**	1.76	0.062 (*)	1.08	0.055	0.27

**^1^** T = found in traces **^2^** Main compounds are shown in bold writing (*) = Highest measured value of the given compound, when compared to the other samples.

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
