# Peer review of "Development and Validation of an HPLC Method for the Quantitative Analysis of Bromophenolic Compounds in the Red Alga Vertebrata lanosa"

_marinedrugs, 2019, doi:10.3390/md17120675_

Round 1
Reviewer 1 Report
marinedrugs-638132-peer-review-v1 Revision This manuscript on bromophenols of the red algae Vertebrata woolly is relevant to this area of knowledge (Marine Drugs), as these chemical compounds have a number of bioactivities, such as antimicrobial, antitumor, antidiabetic and antithrombotic, among others. The text is fluid and well written. I am of the opinion that the manuscript should be accepted for publication after the following corrections have been made. Corrections needed line 4 - ... Vertebrata lanosaline 20 - In the abstract it is not necessary to include with the name of the species (Vertebrata lanosa) the respective authority ((L.) T.A. Christensen), because when it first appears in the introduction the name of the species includes the name of the respective authority.
line 33 - ... against Staphylococcus aureus and Escherichia coli bacteria.
line 44 - ... Rytiphlaea
line 162 - ... (sample Vetebrata lanosa, Brittany, France 2018);
line 165 - ... in Brittany (France)
line 240 - ... Laurencia dendroidea J. Agardh (J. Agardh is not in italics)
lines 248/267 - Put the text in normal letters and not in bold
line 272 - Confirm, please, "GenBank accession number"
line 378 - "Staphylococcus aureus" and "Escherichia coli" in italics, please
Author Response
Dear Reviewer, thank you very much for your valuable comments:
All formal changes were adopted, i.e.line 19 - the respective authority ((L.) T.A. Christensen) was removed
line 33 - the term “bacteria” was added
line 46 – spelling was corrected
line 171 – the sample is specified exactly now
line 175 – the statement was changed for more clarity
line 277 - J. Agardh is not in italics
lines 248/267 – in our version there were no bold letters, except that the compounds are abbreviated by numbers in bold
Line 416 – species names are in italics now
Line 311 – the correct GenBank accession number was added as suggested
Reviewer 2 Report
The manuscript by Hofer et al., describe the isolation of seven bromophenols from the red alga V. Lanosa including one new bromophenol named Vertebratol. The structure elucidation of the new compound is reasonable and the description is convincing. In addition, the authors have optimized HPLC gradient using the seven bromophenols as standard to determine the occurrence of bromophenols in V. Lanosa. The antibacterial activity of the isolated compounds were tested using agar diffusion tests. Three compounds showed weak activity against E.coli and S. aureus.
In order to rationalize such approach some point need to be clarified such as:
- The authors used only the compounds they obtained from V. Lanosa as standards, are they the most important bromophenols from the red algae? Why other compounds (if commercially available) were not included in the study? Only extracts of V. Lanosa were tested to validate the method, in order to validate the method as general procedure to determine the bromophenols in red algae, extracts from other red algae must be screened.
- The authors mentioned in page 2, line 70: that ‘’their study will help to get greater insight in to the occurrence and the role of these compounds in algae’’, nevertheless, they mentioned later (page 10, line 243) that they cannot deduce any environmental parameters, which may determine the occurrence of bromophenols in V. Lanosa. This is confusing and need to be adjusted.
- It would be more interesting to the readers to provide explanation about the ecological importance of the bromophenols rather than their bioactivity, as mentioned by the authors.
- The scafold of the isolated compounds lack the novelity are common in red algae.
- The authos missed some relevant refernces about the bromophenols in marine algae, e.g. Ming liu et al. Marine Drugs 2011, 9(7):1273-92, DOI:10.3390/md9071273; Ana Jesus et al., Mar. Drugs 2019, 17(2), 73; https://doi.org/10.3390/md17020073
- Finally, the authors should address these points and rewrite the manuscript in order to consider the approach as a general approach for detecting the bromophenols in red algae and consider the manuscript for publication in Marine Drugs, otherwise the novelty of the results are not up to the standard and the high IF of the journal.
Author Response
Dear Reviewer, thank you for your comments:
As stated in the manuscripts title, this study focused on the determination of bromophenols in one particular species (Vertebrata lanosa), for which no adequate analytical method was available till date. Accordingly, “only” seven compounds from this species were isolated, structurally elucidated and used for method development. We never claimed that our method is generally applicable for the quantification of the entire class of bromophenols in red algae, and most likely no single method is capable of achieving that. However, in the course of our investigations we analyzed 19 other species of macroalgae and one lichen using our method. Peaks that showed matching retention times with those of the standards were further investigated with HPLC-MS to see if they exhibited the masses of the according standards. The results showed that the seven bromophenols were indeed very characteristic for Vertebrata lanosa, where they are present in fairly high concentrations. They could not be found in any of the other samples. (Respective information was added to the manuscript. Table of the 20 species can be found in the supplement.) We agree on the comment that the last sentence of introduction is possibly exaggerated, because our data does not allow a clear conclusion regarding the role of bromophenols in this particular alga. Thus, the sentence was removed. As stated in introduction already, the bioactivity of bromophenols is better understood than their ecological role. The number of samples investigated in this study won’t permit meaningful conclusions concerning the latter; however, the HPLC method presented is a key requirement to do so. As a consequence, meaningful studies on the ecological role of bromophenols in lanosa will be possible for the first time. We cannot agree on the comment that the isolated compounds lack novelty, because one of them is an overall natural product. Thank you for informing us about those two very nicely written articles. The review you mentioned had already used for the writing of the manuscript. See reference 4: Liu M.; Hansen P.E.; Lin X. Bromophenols in marine algae and their bioactivities. Mar. Drugs 2011, 9, 1273-1292. The other one was added (see reference 27). Concerning the last comment there is a big discrepancy between the reviewers, because reviewers 1 & 2 considered this study relevant for this area of research, as well as interesting and innovative in analytical terms.Reviewer 3 Report
dear Authors, dear Editor
draft article “… bromophenolic Compounds in The Red Alga Vertebrata Lanosa marinedrugs-638132” describes a separation by HPLC of seven phenolic brominated substances from algae and their quantitative determination with UV detection. The merit of this approach is the low cost of the equipment and the caveat on transfer from UV to ESI-MS (change of acid from TFA to formic).
It is none of a reviewer’s business, but the text is untidily formatted (I have problems with the template myself!), so a revised draft should amend this. In addition (and this is not the Authors’ fault) I definitely disagree with placing the Materials and Methods at the end. A reader cannot jump pages to find information that allow understanding the results. For the same reason, as a reviewer, I make silly comments while reading top-down, and I have to strike them when I arrive to the M&M section at the end.
I have issues on the presentation of analytical performance, if this article aims at establishing a method for quantitative analysis.
L.78: pseudo 79 molecular ion peak cluster. Please, show me a “pseudo-molecule”, and I’ll afford you a pseudo-molecular ion: this is a joke from a senior mass spectrometrist who argued against this pseudo-way of naming molecular precursors. Yours is a protonated or de-protonated molecule. By the way, why didn’t you record and publish MS/MS spectra of the compounds?
Table 2. I will not agree on LoD/LoQ determined from extrapolation 100 times lower than that of the lowest calibrator. I suggest that you obtain calibration lines with appropriately lower-level calibrators close to the expected LoD/LoQ. That stated in L. 366-368 does not apply.
Accuracy and precision as well are not reported correctly: I would expect to find measured vs. theoretical actual values and percent displacement for accuracy; for precision the CV% and for all the number of replicates.
Reference to the determination in the six samples (Table 3): opt for microg or mg, consistently (I suggest microg). Invert tables 3 and 4 or, better, put M&M as section 2 and put Table 4 there.
If you will need to feed results to multivariate analysis, e.g., to discriminate species or to investigate environmental effects on biosynthesis or for other purposes, accuracy and precision are mainstay. You may mix a representative of all analyzed extracts and use it for QC.
L.221: good of you to note! Many a mass spectrometrist would ask you to compare TFA and formic acid also in the HPLC-UV, to see whether there is a difference in separation power.
In the Discussion, six samples are a bit too few to embark in such a detailed discussion on the different levels of the metabolites: I am not aware of the literature, but if there is more, you may wish to tabulate results in the cited articles.
Why is Lines 248-267 in bold?
I strongly suggest that the text is reshuffled in the conventional order, with M&M as section 2, then analytical performance is reported in the “traditional” way.
Best regards
Author Response
Dear Reviewer, thank you for your valauble comments:
We do understand your concerns regarding the layout of the manuscript, which might be unusual for some. However, these are the guidelines of the journal, so we did not change the order of the sections. Thank you for your comment. In order to avoid any confusion we replaced the term “pseudo-molecule” with the more precise expression “typical isotopic pattern for dibrominated molecules”. Our work aimed to develop a simple and cost-efficient method for the identification and quantification of bromophenolic compounds, which could potentially be utilized in any laboratory equipped with an HPLC. HPLC-MS was used to ensure that the peaks in the samples actually corresponded to the standards, not only in terms of retention times but also the correct masses. Furthermore, HPLC-MS was also beneficial during the isolation process of the standards to identify brominated compounds in the extract. MS/MS spectra were not necessary to elucidate the structures of the compounds or to assign the peaks, as all structures were unambiguously elucidated with 1D and 2D NMR analysis. Thank you for your valuable comment. LOD values, however, were only up to 14 (not 100) times lower than the concentrations of the lowest calibrator, which is not ideal either. In order to narrow the gap we additionally analyzed two further dilutions of the standards and included this data in method validation, i.e. linearity range as well as calculation of LOD and LOQ values (see the revised Table 2). The respective values changed slightly, and therefore were corrected in the manuscript; this also applies to some of the quantitative results. The presentation of accuracy and precision results was changed as suggested (also see revised table 2). As suggested, mg/g was changed to µg/mg, and tables 3 and 4 were inverted. Thank you for your suggestion regarding the possibility to use the data for multivariate analysis. However, the aim of this work was solely to develop an HPLC method for the determination of bromophenolic compounds in this particular alga. The analysis of six samples should show that the method works in practical terms. We will consider this point for pursuing investigations. There were indeed differences in separation power depending on the type of acid used. A UV-chromatogram using HPLC-MS conditions is shown in Supplementary Figure 2 already. We are aware that the results of six samples are not sufficient to deduce any general statements of ecological or chemosystematic relevance. However, there are some possible trends and Marine Drugs is not a purely analytical journal. Thus, we did not modify the discussion of results. Like for reviewer 1, we did not submit lines 248-267 in bold. Maybe this occurred while the format of the manuscript was changed during submission.Round 2
Reviewer 2 Report
I have checked the revised version of the manuscript, the authors have impoved the quality of the paper. Therefore, I recommend publishing in Marine Drugs.
Reviewer 3 Report
Dear Authors, dear Editor,
I have reviewed the revised version “marinedrugs-638132-peer-review-v2”. The Authors took care of improving the presentation and presented sound arguments for their choices. I suggest that this version is adequate for publishing in Marine Drugs.
Best regards.